# ATTACKING FOR INSPECTION AND INSTRUCTION: AT-TACK TECHNIQUES CAN AID IN INTERPRETABILITY

## ABSTRACT

This study investigates a data-centric self-explaining framework constructed with a cooperative game, where a generator first extracts the most informative segment (i.e., rationale) from raw input, and a subsequent predictor utilizes the selected subset for its input. The generator and predictor are trained collaboratively to maximize prediction accuracy. In this paper, we first uncover a potential caveat: such a cooperative game could unintentionally introduce a sampling bias during rationale extraction. Specifically, the generator might inadvertently create an incorrect correlation between the selected rationale candidate and the label, even when they are semantically unrelated in the original dataset. Subsequently, we elucidate the origins of this bias using both detailed theoretical analysis and empirical evidence. Our findings suggest a direction for inspecting these correlations through attacks, based on which we further introduce an instruction to prevent the predictor from learning the correlations. Through experiments on six text classification datasets and one graph classification dataset using three network architectures (GRUs, BERT, and GCN), we show that our attack-inspired method not only outperforms the vanilla rationalization method but also beats several recent competitive methods. We also compare our method against a representative LLM (llama-3.1-8b-instruct), and demonstrate that our approach achieves comparable results, sometimes even surpassing it. Code: https://anonymous.4open.science/r/A2I-A700.

## 1 INTRODUCTION

With the success of deep learning, there are growing concerns over the model interpretability. Exploring the theory and technique of interpretable machine learning frameworks is of immense importance in addressing a myriad of issues. For instance, XAI techniques can aid in detecting model discrimination (fairness) (Pradhan et al., 2022), identifying backdoor attacks (security) (Li et al., 2022), and revealing potential failure cases (robustness) (Chen et al., 2022), among others. Post-hoc explanations, which are trained separately from the prediction process, may not faithfully represent an agent's decision, despite appearing plausible (Lipton, 2018). In contrast to post-hoc methods, ante-hoc (or self-explaining) techniques typically offer increased transparency (Lipton, 2018) and faithfulness (Yu et al., 2021), as the prediction is made based on the explanation itself. There is a stream of research that has exposed the unreliability of post-hoc explanations and called for self-explanatory methods (Rudin, 2019; Ghassemi et al., 2021; Ren et al., 2024).

In this study, our primary focus is on investigating a general model-agnostic self-explaining framework called Rationalizing Neural Predictions (RNP, also known as rationalization) (Lei et al., 2016), which with its variants has become one of the mainstream methods to facilitate the interpretability of NLP models (Sha et al., 2021; Yu et al., 2021; Liu et al., 2022; 2023; Storek et al., 2023), and also holds the potential to be applied to image classification (Yuan et al., 2022) and graph neural networks (Luo et al., 2020). RNP utilizes a cooperative game involving a generator and a predictor. This game is designed with a focus on "data-centric" (i.e., it is to explain the connection between a text and the (model-agnostic) task label, rather than explaining the output of a specific model) feature importance. The generator first identifies the most informative part of the input, termed the rationale. Subsequently, the rationale is transmitted to the predictor to make predictions, as illustrated in Figure 1. The generator and predictor are trained cooperatively to maximize prediction accuracy.

Figure 1: The standard rationalization framework RNP. The task in this figure is binary sentiment classification about hotels' service. $X, Z, \hat{Y}, Y$ represent the input, the selected rationale candidate, the prediction, and the classification label. $M$ is a sequence of binary masks. $\theta_g, \theta_p$ are the parameters of the generator and the predictor.

Apart from its use for interpretability, some recent studies find that rationalization can also serve as a method for data cleaning. The extracted $(Z, Y)$ pairs can act as a new dataset, and trained with such a cleaned dataset, a predictor may be more robust (Chen et al., 2022) and generalizable (Wu et al., 2022; Gui et al., 2023), thanks to the removal of task-irrelevant, harmful information.

Our research starts with a special empirical observation. We first observe that, even if we remove "maximizing the prediction accuracy" from the generator's objective (thus it selects some random noise), the predictor can still be trained to get very high accuracy with these randomly selected spurious rationales (the orange line in Figure 4(a) of §4.1). This phenomenon then leads to a trust concern: whether the extracted rationale is really responsible for the label in the original dataset (i.e., although the extracted rationale is considered faithful to the model's prediction by previous research, is it faithful to the model-agnostic dataset?). This problem is important because explanations should also be aligned with their social attribution (Jacovi & Goldberg, 2020; 2021).

> **Task:** Binary sentiment classification
> **Label** (about the beer's appearance): Positive. **Prediction**: Positive.
> **Input:** a - murky , semi-opaque honey . low head . s -earthy.
> plantains , pineapple rind , apricot t - earthy hay and pepper . touch
> or orange . cilantro . honey . very saison-like . m - medium body .
> nice carbonation . balanced semi-dry finish . o - nice flavor profile .
> **Rationale selected by RNP**: ["."]

Figure 2: A cherry-picked example of the generator-added spurious correlation. The underlined text is human-annotated rationale. The text in red is the rationale selected by RNP. *Example 1*: from a positive input $X^1$ with a label 1, the generator selects a rationale $Z^1$ that includes the pattern "."; and for a negative input $X^0$ with a label 0, the generator selects a rationale $Z^0$ that does not include".". And subsequently, the predictor considers the presence or absence of "." as an indicative feature for positive classification.

We then shed light on the source of this problem. Typically, we call a pattern $T$ is **t**rivial if it is independent with $Y$ in the original dataset: $P(Y|T) = P(Y)$. However, due to the potential bias of the generator's sampling, $T$ can be correlated with $Y$ in the sampled $(Z, Y)$ pairs. Figure 2 provides a (cherry-picked) practical example of it.

We further explore the origins of this issue and discover that it stems from an approximation that was overlooked in previous research: taking a series of $(Y, Z)$ pairs sampled by the generator as an approximation of $P(Y, Z)$ (while it should actually be $P(Y, Z|g)$, and note that $Y \perp\!\!\!\perp Z \not\Rightarrow Y \perp\!\!\!\perp Z|g$). In fact, this problem can be seen as a type of spurious correlation. But notably, the perspective of this paper is totally different from the traditional causality research for spurious correlations. Existing research on causality has primarily focused on spurious correlations inherent in the dataset. However, our research investigates a further question: *if the dataset itself is clean and lacks spurious correlations, could the selection process of the generator introduce additional spurious correlations?*

This study tries to address this kind of correlations with two steps: inspection and instruction. We first theoretically show that if a predictor classifies based on a trivial pattern $T$ that is associated with the category label $Y$ due to the sampling of the generator, we can always find an attacker to inspect the trivial pattern. Then, to prevent the predictor from learning such a correlation (which would make the generator further enhance it), we manually adjust the distribution of the trivial pattern from $P(Y|T, g)$ to $P(Y)$ (in fact, it should be $P(Y|T)$, but we have $P(Y|T) = P(Y)$ for the attacker identified trivial pattern $T$) to provide instructions that enable the predictor to learn the correct information, thereby giving the generator the correct feedback. We provide a toy example in Appendix A.2 to give readers a more intuitive understanding of our method.

In summary, our contributions include: (a) We identify a new type of spurious correlation, and we systematically analyze how it can arise in a clean dataset with both theoretical support and empirical

verification. (b) A practical solution. We design an attacker to both inspect whether the predictor has learnt from the spurious correlation and instruct the predictor not to learn from it. (c) We design various experiments to verify the existence of the generator added spurious correlation, the effectiveness of the inspection, and the effectiveness of the instruction. Besides, the attack based inspection and instruction is model-agnostic, so we conduct it on top of both the standard RNP and an advanced variant FR (Liu et al., 2022), and all get improved performance. (d) Research on attacks is primarily used to inspire defense methods and ensure model security. However, our work demonstrates that attacks can also aid in interpretability, representing an important attempt to bridge the security community and the XAI community.

## 2 RELATED WORK

**Rationalization**. The basic cooperative framework of rationalization named RNP (Lei et al., 2016) is flexible and offers a unique advantage: certification of exclusion, which means any unselected input is guaranteed to have no contribution to prediction, making it important to the NLP community (Yu et al., 2021). Based on it, many methods have been proposed to improve RNP from different aspects. Bao et al. (2018) used Gumbel-softmax to do the reparameterization for binarized selection. Bastings et al. (2019) replaced the Bernoulli sampling distributions with rectified Kumaraswamy distributions. Jain et al. (2020) disconnected the training regimes of the generator and predictor networks using a saliency threshold. Paranjape et al. (2020) imposed a discrete bottleneck objective to balance the task performance and the rationale length. **?** proposed a benchmark that can be used for supervised rationale extraction. Inter_RAT (Yue et al., 2023) tried to use backdoor adjustment to alleviate the spurious correlations in the raw dataset. Havrylov et al. (2019) cooperatively trained the models with continuous and discrete optimisation schemes. (Hase et al., 2020) explored better metrics for evaluation. (Rajagopal et al., 2021) used phrase-based concepts to conduct a self-explaining model. Other methods like data augmentation with pretrained models (Plyler et al., 2021), training with human-annotated rationales (Chan et al., 2022), injecting noise to the selected rationales (Storek et al., 2023), have also been tried.

Prior to our work, a series of studies had observed a phenomenon termed degeneration, whose origin can also be attributed to the spurious correlation we investigate in this study. Degeneration means that, the predictor is too powerful to recognize any trivial patterns that are distinguishable in rationales with opposite labels. As a result, the generator may collude with the predictor to select the trivial patterns rather than the true semantics as the rationales (Yu et al., 2019). Previous methods seek to regularize the model using supplementary modules which have access to the information of the full text (Yu et al., 2019; Huang et al., 2021; Yu et al., 2021; Liu et al., 2022) such that the generator and the predictor will not overfit uninformative rationales. 3PLAYER (Yu et al., 2019) tries to squeeze the informative texts from the unselected parts to produce comprehensive rationales. DMR (Huang et al., 2021) tries to align the distributions of rationale with the full input text in both the output space and feature space. A2R (Yu et al., 2021) endows the predictor with the information of full text by introducing a soft rationale. FR (Liu et al., 2022) folds the two players to regularize the predictor with the generator (as the generator can view the raw input) by sharing a unified encoder. Among them, FR achieves the strongest improvements on addressing degeneration, and will be included in our baselines. However, although these methods have been proposed to fix the observed problem, the origin of this problem is not well explored. Sometimes they can still fail. For example, Zheng et al. (2022) argued with both philosophical perspectives and empirical evidence that the degeneration problem is much more complex than we used to think and some of the above methods cannot promise no-degeneration. In fact, this phenomenon is similar to what we discuss and can also be seen as one of the problems stems from taking $P(Y, Z|g)$ as $P(Y, Z)$, highlighting the importance of rectifying the bias in approximating $P(Y, Z|g)$ as $P(Y, Z)$.

We also briefly discuss the potential impact of rationalization in the era of LLMs in Appendix A.1. We compare our method against a representative LLM (llama-3.1-8b-instruct) in Appendix A.6.

## 3 DEFINITION OF THE RATIONALIZATION TASK

**Notations**. Unless otherwise specified, uppercase letters represent random variables, while lowercase letters correspond to their values. For simplicity, we do not distinguish between vectors and scalars. We consider the classification task. We have a classification dataset $\mathcal{D}$, which can be seen

as a collection of samples drawn from the true data distribution $P(X, Y)$. $X = X_{1:l}$ is the input text sequence with a length of $l$, and $Y$ represent the classes in the dataset (note that a discrete label can also seen as representing a distribution like [0,1]). By enumerating $X$, we can get $P(Y|X)$, which is the distribution that a normal non-interpretable classifier working on $\mathcal{D}$ needs to approximate. Rationalization consists of a generator $f_g(\cdot)$ (or $g$ for conciseness) and a predictor $f_p(\cdot)$, with $\theta_g, \theta_p$ being their parameters.

For $(X, Y) \sim \mathcal{D}$, the generator first outputs a sequence of binary mask $M = f_g(X) = M_{1:l} \in \{0, 1\}^l$ (in practice, the generator first outputs a Bernoulli distribution for each token and the mask for each token is independently sampled using gumbel-softmax). Then, it forms the rationale candidate $Z$ by the element-wise product:

$$Z = M \odot X = [M_1 X_1, \cdots, M_l X_l]. \tag{1}$$

To simplify the notation, we denote $f_g(X)$ as $Z$ in the following sections, i.e., $f_g(X) = Z$.

We consider that $X$ consists of a set of variables $\{T_1, \cdots, T_n, R\}$, where $R$ denotes the real rationale (e.g., sentiment tendency for sentiment classification) for task label $Y$, and $T_1, \cdots, T_n$ are some **t**rivial patterns independent with $Y$. And we select one of $\{T_1, \cdots, T_n, R\}$ to be $Z$. Note that $Z$ is not a separate variable but a proxy for any variable within $X$. Till now, we get a set of $(Z, Y)$ samples denoted as $\mathcal{D}_{\mathcal{Z}}$. Previous research simply thinks $\mathcal{D}_{\mathcal{Z}}$ is collected from $P(Z, Y)$. By enumerating $Z$ in $\mathcal{D}_{\mathcal{Z}}$, they get $P(Y|Z)$. Then, they attempt to identify the rationale by maximizing the mutual information:

$$Z^* = \underset{Z \in \{T_1, \cdots, T_n, R\}}{\arg\max} I(Y; Z) = \underset{Z \in \{T_1, \cdots, T_n, R\}}{\arg\max} \left( H(Y) - H(Y|Z) \right) = \underset{Z \in \{T_1, \cdots, T_n, R\}}{\arg\min} H(Y|Z). \tag{2}$$

In practice, the entropy $H(Y|Z)$ is commonly approximated by the minimum cross-entropy $\min_{\theta_p} H_c(Y, \hat{Y}|Z)$, with $\hat{Y} = f_p(Z)$ representing the output of the predictor (note that the minimum cross-entropy is equal to the entropy, Appendix B.3). Replacing $Z$ with $f_g(X)$, the generator and the predictor are trained cooperatively:

$$\min_{\theta_g, \theta_p} H_c(Y, f_p(f_g(X))|f_g(X)), \ s.t., \ (X, Y) \sim \mathcal{D}. \tag{3}$$

**Compactness and coherence**. To make the selected rationales human-intelligible, previous methods usually constrains the rationales by compact and coherent regularization terms. In this paper, we use the most widely used constraints provided by Chang et al. (2019):

$$\Omega(M) = \lambda_1 \left| \frac{\|M\|_1}{l} - s \right| + \lambda_2 \sum_{t=2}^{l} \left| M_t - M_{t-1} \right|. \tag{4}$$

The first term encourages that the percentage of the tokens being selected as rationales is close to a pre-defined level $s$. The second term encourages the rationales to be coherent. We adopt both compactness and coherence regularizers to the generator to make the rationales human-intelligible. We apply a compactness regularizer term to the attacker to make the attack rationale more similar to the original rationale, thus making it easier to deceive the predictor. However, we do not employ a coherence regularizer on it because we think trivial patterns are often discontinuous.

## 4 MOTIVATION AND METHOD

**Notation.** For the sake of exposition, let us take the example of binary sentiment classification. We denote $X^1$ and $X^0$ as input texts with label $Y = 1$ and $Y = 0$, respectively. $Z$ and $Z_A$ represent the rationale candidates selected by the generator and the attacker, respectively. Note that they are not separate variables but a proxy for any variables within $X$. Sometimes we use $Z$ and the variable represented by $Z$ interchangeably. $T$ is a proxy for any variables within $\{T_1, \cdots, T_n\}$ (defined in §3).

### 4.1 CAUSE OF THE SPURIOUS CORRELATION

**How do trivial patterns correlate with $Y$?** Although considering $\mathcal{D}_{\mathcal{Z}}$ as an approximation of $P(Z, Y)$ seems to be a simple and practical way and is inherited by all the previous methods (§3),

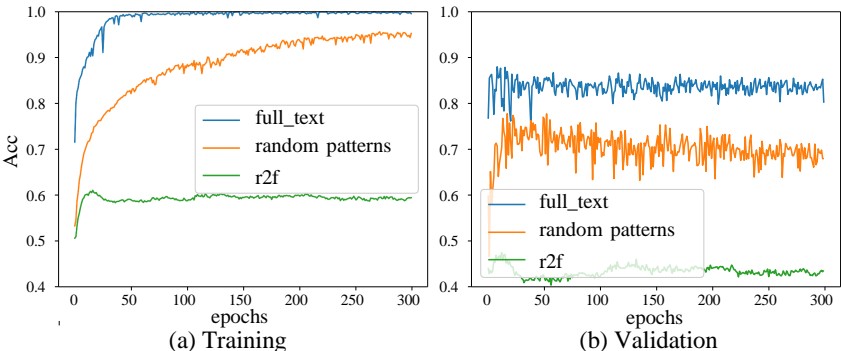

(a) Training   (b) Validation

Figure 4: Experiments on the Beer-Aroma dataset: "full text": a predictor trained using the full texts. "random patterns": a predictor trained with randomly selected patterns. "r2f": feeding the random patterns to the predictor that was trained using the full texts.

it will sometimes results in some problems. In fact, the sampling process of $Z$ is conditioned on a generator $g$ with specific parameters $\theta_g$. So we can only get $P(Z, Y|g)$ and $P(Y|Z, g)$ rather than $P(Z, Y)$ and $P(Y|Z)$. Note that independent doesn't lead to conditional independent: $Y \perp\!\!\!\perp Z \not\Rightarrow Y \perp\!\!\!\perp Z|g$. That is to say, some uninformative $Z$ (like those $T_1, \cdots, T_n$) might initially be independent with $Y$ and maintain zero mutual information with $Y$. But sampled by $g$, any trivial patterns may get correlated with $Y$ and get increased mutual information, thus can be used as (incorrect) indicative features for classification.

What's more, the training process may even enhance the sampling bias further. For example, we consider $T_1$ is selected as $Z$, then the updating of the generator is $\theta'_g = h(\theta_g, T_1, Y)$ ($h$ denotes the backpropagation process), and this structural function corresponds to a small local of a causal graph shown in Figure 3. We originally have $Y \perp\!\!\!\perp T_1$. But in this graph, we have $Y \not\perp\!\!\!\perp T_1|G$. That's to say, any trivial patterns hold the potential to be associated with $Y$ through the influence of the generator.

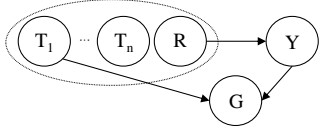

Figure 3: A local of the causal graph for the generator's updating process. Dash cycle means $X$ consists of a set of variables.

Consider a situation where $Z = T$ is a trivial pattern independent with $Y$ (i.e., $P(Y = 1|T) = P(Y = 1) = 0.5 = P(Y = 0) = P(Y = 0|T)$ and $T \in \{t_+, t_-\}$). Influenced by the generator $g$, $T = t_+$ might co-occur more frequently with $Y = 1$ and can be viewed as an indicator for the positive class ($T = t_-$ is similar):

$$\begin{cases} P(Y = 1|Z = t_+, g) > P(Y = 1) \\ P(Y = 0|Z = t_+, g) < P(Y = 0). \end{cases} \tag{5}$$

Example 1 in Figure 2 of §1 also provides an intuition for the above analysis.

**Empirical support**. The above motivation is inspired by some practical observations. We present three types of prediction accuracies for a binary sentiment classification task (about the beer's aroma) in Figure 4: ① A predictor trained with the full input text. ② A predictor trained with randomly selected patterns. For the generator, we remove the other objectives and only train it with the sparsity constraints (Equation 4). That is to say, the generator is trained to randomly select $10\%$ of the input text, and the predictor is then trained to classify using these randomly selected texts. ③ We use the randomly selected texts from ② to feed the predictor trained in ①.

From Figure 4(a), we observe that even with the randomly selected patterns (i.e., patterns unlikely to contain real rationales), the predictor can still achieve a very high prediction accuracy (represented by the orange line, approximately $95\%$). This accuracy is close to that of the classifier trained with the full texts. A follow-up question is: Does this strange result stem from the fact that the $10\%$ randomly selected patterns already contain enough sentiment inclination for classification? The answer is no. Consider the green line, which represents the outcome when we feed the randomly selected texts to the well-trained predictor denoted by the blue line. We observe that the green line indicates a significantly lower accuracy (about $58\%$), implying that the randomly selected patterns contain only minimal sentiment information. Thus, the orange predictor incorrectly treats certain

randomly selected trivial patterns as indicative features. Moreover, the orange predictor does not generalize well to the validation set (Figure 4(b)), due to the fact that simple trivial patterns can more easily lead to overfitting (Pagliardini et al., 2023).

We provide more evidence of the existence of such spurious correlations in practical scenarios from another perspective by demonstrating the attack success rate in §5.1.

## 4.2 THE PROPOSED METHOD

For the sake of clarity in reading, we first present our approach and subsequently expound on the principles underlying it.

Figure 5 shows the architecture of our method. For a data point $(X, Y)$ in a n-class classification task, the over all objective of our model ( $f_p, f_g, f_a$ represent the predictor, the generator, and the attacker, with $\theta_p, \theta_g, \theta_a$ being their parameters) is:

$$\textbf{attacker}: \min_{\theta_a} H_c(Y_A, f_p(f_a(X))|f_a(X)), \tag{6}$$

$$\textbf{gen\&pred}: \min_{\theta_g, \theta_p} H_c(Y, f_p(f_g(X))|f_g(X)) + \min_{\theta_p} H_c([1/n, \cdots, 1/n], f_p(f_a(X)|f_a(X)) \tag{7}$$

$$s.t.\ Y_A = \text{randint}(0, n)\&Y_A \neq Y. \tag{8}$$

$Y_A$ represents the class to be attacked. We randomly select a class for each attack to create a balanced attack for each class. $[1/n, \cdots, 1/n]$ represents the distribution of $P(Y)$ in the raw dataset. $\min_{\theta_p} H_c([1/n, \cdots, 1/n], f_p(f_a(X)|f_a(X))$ means we rectify the sampled distribution of $P(Y|Z_A, a)$ to $P(Y)$ and ask the predictor to learn that $Z_A$ is not correlated with $Y$. In binary classification, we have $Y_A = 1 - Y$ and $1/n = 0.5$.

During training, (7) and (6) are alternated. The practical implementation details with Pytorch are in Appendix A.3. The overall mechanism of the model is as follows: (6) inspects trivial patterns ($f_a(X)$) from $X$. The second term of (7) is the instruction that prevents the predictor from learning the trivial patterns by classifying them as random noise. A well instructed predictor is then able to give good feedback to the generator's selection. And the first term of (7) is the normal RNP. The reason why the attacker constructed in this manner can detect trivial pat-

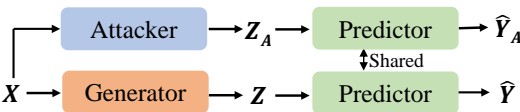

Figure 5: The architecture of attacking for inspection and instruction. We name it **A**ttack to **I**nspection and **I**nstruction (A2I). $Z, Z_A$ represent the selected rationale candidate and the attack rationale. $\hat{Y}, \hat{Y}_A$ represent the normal prediction and the attack result.

terns will be elucidated in §4.3. We also use a toy example in Appendix A.2 to provide an intuitive understanding. At the end of §4.3, we also discuss how our method will work in the situation where the generator and the predictor cooperate correctly on real rationales rather than trivial patterns.

## 4.3 UNDERLYING PRINCIPLES

**Attack as inspection**. Following the above settings for $Z = T$ and $I(Y; T) = 0$ in §4.1, we will show how the trivial patterns learned by the predictor can be inspected through attack. Corresponding to (5), if the attack generator can be constructed in any way (i.e., has infinite expressiveness), then we can always find an attack generator $g_a$ which extracts $Z_A$ from $X$, such that

$$\begin{cases} P(Y = 1|Z_A = t_+, g_a) < P(Y = 1) \\ P(Y = 0|Z_A = t_+, g_a) > P(Y = 0). \end{cases} \tag{9}$$

Appendix B.1 shows the detailed derivation for the reason why we can find such a $g_a$. Equation (9) is the opposite of (5), and it means that under condition $g_a$, $T = t_+$ now becomes a negative class indicator, which is exactly the opposite situation under condition $g$. Here is the intuitive understanding of the attack. Corresponding to the punctuation pattern example mentioned in Figure 2 of §1. The generator $g$ selects $Z =$ "." from $X^1$. And the predictor has learnt to predict "." as positive. We can employ an attacker $g_a$ which selects $Z_A =$ "." from $X^0$ (whose class label is negative) such that $Z_A$ can also be classified as positive. Similarly, the attacker can find $Z_A =$ "," from $X^1$ to be classified

as negative. So, the overall objective of the attacker is to select those $Z_A$ that can be classified to the opposite class by the predictor.

Formally, the objective of the attacker is

$$\min_{\theta_a} H_c(1 - Y, f_p(f_a(X))|f_a(X)). \tag{10}$$

Till now, we have demonstrated that an attacker can identify uninformative trivial patterns and classify them into the opposite class. Then we begin to instruct the predictor to not learn from the trivial patterns (whether the attacker will select real rationales is discussed at the end of this section).

**Attack as instruction.** When the spurious correlation occurs, the attacker $g_a$ consistently chooses a $Z_A$ that is a label-independent trivial pattern. For a competent predictor $p$ that discerns the authentic rationale, $Z_A$ resembles noise independent with $Y$, ensuring its classification remains random without any leanings to a specific label. Thus, we introduce an extra instruction to the predictor:

$$\min_{\theta_p} H_c([0.5, 0.5], f_p(Z_A)), s.t., \ Z_A = f_a(X), \ (X, Y) \sim \mathcal{D}. \tag{11}$$

That is to say, although we cannot promise the independence between $Z_A$ and $Y$ under the generator's conditional sampling, we can make $Z_A \perp\!\!\!\perp \hat{Y}$ through the predictor's prediction.

**The situation of a text $X$ contains both positive and negative sentiments**. Here we consider $Z = R$, which is the true rationale based on which the label $Y$ is assigned to $X$. We denote $R = r_+, R = r_-$ as positive and negative indicators, respectively. The question we want to discuss now is, if the generator and the predictor cooperates well on real rationales, what will happen if $X$ contains both positive and negative sentiments?

The first glance might be that, both the generator and the attacker choose the true (but opposite) sentiment rationales, thereby leading to the predictor in (7) being unable to make the right prediction. But in practice, the predictor can overcome this obstacle. Consider an intuitive assumption:

**Assumption 1.** *The positive rationale $r_+$ appears more often in positive texts than in negative ones:* $P(r_+|Y = 1) \geq P(r_+|Y = 0)$.

This assumption stems from that we can always find $r_+$ in $X^1$, but sometimes not in $X^0$. If Assumption 1 holds, we can easily prove (please refer to Appendix B.2) that the predictor in (7) will still converge to predict $f(r_+)$ as positive with a high confidence ($\geq 0.75$).

## 5 EXPERIMENTS

**Baselines**. We compare our A2I with the standard RNP and several recent representative methods: Inter_RAT (Yue et al., 2023) and CR (Zhang et al., 2023) represent recent causal methods, and FR (Liu et al., 2022) and NIR (Storek et al., 2023) represent recent methods designed to deal with degeneration. All of them have been discussed in §2.

**Datasets**. We first follow FR to examine on three datasets from BeerAdvocate benchmark (McAuley et al., 2012): Beer-Appearance, Beer-Aroma, Beer-Palate, and three datasets from HotelReview benchmark (Wang et al., 2010): Hotel-Location, Hotel-Service, Hotel-Cleanliness. Among them, the three beer-related datasets are used by nearly all of previous research in the field of rationalization. We also use a graph rationalization dataset, BA2Motifs (Ying et al., 2019), to verify generalizability. These datasets include human-annotated rationales in their test sets to facilitate objective comparison between different methods. More details about the datasets are in Appendix A.4.

**Metrics**. Our findings in this paper suggest that the prediction performance is not a good metric for the models' effectiveness. Following Inter_RAT and FR, we mainly focus on the rationale quality, which is measured by the overlap between model-selected tokens and human-annotated rationales. The terms $P, R, F1$ denote precision, recall, and $F1$ score respectively. The term $S$ represents the average sparsity of the selected rationales, that is, the percentage of selected tokens in relation to the full text. $Acc$ stands for the predictive accuracy.

**Implementation details**. The generator, predictor, and attacker all are composed of an encoder (RNN/Transformer/GCN) and a linear layer. We use three kinds of encoders: GRUs (following

Table 1: Results on datasets from the BeerAdvocate benchmark. Inter_RAT: Yue et al. (2023), NIR: Storek et al. (2023), FR: Liu et al. (2022). We follow Inter_RAT to set $S \approx 10\%, 20\%, 30\%$.

| Datasets | | Beer-Appearance | | | | | Beer-Aroma | | | | | Beer-Palate | | | | |
|---|---|---|---|---|---|---|---|---|---|---|---|---|---|---|---|---|
| Methods | | S | Acc | P | R | F1 | S | Acc | P | R | F1 | S | Acc | P | R | F1 |
| Comparison with standard RNP | | | | | | | | | | | | | | | | |
| $S \approx 10\%$ | RNP | 10.1 | 79.7 | 69.3 | 37.6 | 48.8 | 10.0 | 82.9 | 81.3 | 52.4 | 63.7 | 9.3 | 84.7 | 68.6 | 51.3 | 58.7 |
| | RNP+A2I | 10.8 | 82.8 | 78.3 | 45.8 | **57.8** | 9.8 | 86.3 | 86.0 | 54.3 | **66.6** | 10.9 | 86.6 | 66.3 | 58.2 | **62.0** |
| $S \approx 20\%$ | RNP | 19.8 | 86.3 | 69.8 | 74.6 | 72.1 | 20.7 | 84.5 | 43.6 | 58.1 | 49.8 | 20.1 | 82.6 | 47.6 | 77.0 | 58.8 |
| | RNP+A2I | 20.0 | 87.7 | 73.3 | 79.4 | **76.2** | 19.5 | 85.4 | 49.0 | 61.4 | **54.5** | 19.4 | 86.6 | 49.0 | 76.4 | **59.7** |
| $S \approx 30\%$ | RNP | 30.4 | 84.3 | 52.9 | 86.7 | 65.7 | 30.7 | 81.8 | 39.2 | 77.2 | 52.0 | 30.1 | 87.1 | 29.3 | 71.0 | 41.5 |
| | RNP+A2I | 29.9 | 85.2 | 59.3 | 95.9 | **73.3** | 27.8 | 87.3 | 44.5 | 79.3 | **57.0** | 30.5 | 87.1 | 30.8 | 75.5 | **43.7** |
| Comparison with advanced variants | | | | | | | | | | | | | | | | |
| $S \approx 10\%$ | Inter_RAT | 13.2 | - | 50.0 | 35.7 | 41.6 | 13.8 | - | 64.0 | 56.9 | 60.2 | 13.0 | - | 47.2 | 49.3 | 48.2 |
| | NIR | 10.6 | 78.1 | 77.0 | 44.3 | 56.2 | 10.3 | 86.1 | 74.9 | 49.7 | 59.8 | 11.5 | 84.0 | 48.1 | 44.4 | 46.2 |
| | FR | 11.0 | 82.2 | 68.0 | 40.5 | 50.8 | 9.4 | 86.7 | 85.3 | 51.5 | 64.2 | 9.4 | 84.5 | 70.1 | 52.8 | 60.2 |
| | FR+A2I | 11.3 | 84.6 | 76.0 | 46.5 | **57.7** | 10.0 | 86.9 | 85.7 | 54.8 | **66.9** | 9.7 | 84.8 | 71.4 | 55.8 | **62.6** |
| $S \approx 20\%$ | Inter_RAT | 20.2 | - | 45.8 | 50.4 | 48.0 | 22.0 | - | 47.2 | 67.3 | 55.5 | 20.2 | - | 39.9 | 64.9 | 49.4 |
| | NIR | 20.3 | 81.9 | 70.3 | 77.2 | 73.6 | 19.1 | 87.7 | 61.2 | 75.2 | 67.5 | 19.9 | 83.9 | 37.3 | 59.6 | 45.9 |
| | FR | 19.7 | 87.7 | 77.7 | 82.8 | 80.2 | 20.5 | 90.5 | 61.1 | 80.3 | 69.4 | 19.8 | 86.0 | 42.1 | 67.0 | 51.7 |
| | FR+A2I | 19.8 | 88.7 | 80.0 | 85.6 | **82.7** | 19.4 | 89.7 | 64.2 | 80.0 | **71.2** | 19.2 | 86.0 | 44.2 | 68.2 | **53.7** |
| $S \approx 30\%$ | Inter_RAT | 28.3 | - | 48.6 | 74.9 | 59.0 | 31.5 | - | 37.4 | 76.2 | 50.2 | 29.2 | - | 29.7 | 69.7 | 41.7 |
| | NIR | 29.6 | 84.9 | 59.8 | 95.5 | 73.6 | 30.0 | 82.3 | 38.4 | 73.9 | 50.5 | 29.7 | 84.1 | 22.8 | 54.5 | 32.2 |
| | FR | 30.0 | 90.9 | 58.5 | 94.6 | 72.3 | 31.0 | 83.2 | 40.0 | 79.4 | 53.2 | 29.3 | 84.8 | 28.5 | 67.2 | 40.1 |
| | FR+A2I | 28.8 | 89.7 | 61.3 | 95.3 | **74.6** | 30.9 | 83.2 | 41.4 | 82.2 | **55.1** | 29.1 | 85.1 | 31.6 | 73.8 | **44.2** |

Table 2: Results on datasets from the HotelReview benchmark. We follow FR to set $S \approx 10\%$. *: results from Table 2 of FR.

| Datasets | | Hotel-Location | | | | | Hotel-Service | | | | | Hotel-Cleanliness | | | | |
|---|---|---|---|---|---|---|---|---|---|---|---|---|---|---|---|---|
| Methods | | S | Acc | P | R | F1 | S | Acc | P | R | F1 | S | Acc | P | R | F1 |
| Comparison with standard RNP | | | | | | | | | | | | | | | | |
| $S \approx 10\%$ | RNP* | 8.8 | 97.5 | 46.2 | 48.2 | 47.1 | 11.0 | 97.5 | 34.2 | 32.9 | 33.5 | 10.5 | 96.0 | 29.1 | 34.6 | 31.6 |
| | RNP+A2I | 9.0 | 97.5 | 50.2 | 53.4 | **51.7** | 11.6 | 97.0 | 46.8 | 47.4 | **47.1** | 9.7 | 96.5 | 34.7 | 38.2 | **36.4** |
| Comparison with advanced variants | | | | | | | | | | | | | | | | |
| $S \approx 10\%$ | Inter_RAT | 11.0 | - | 34.7 | 44.8 | 39.1 | 12.5 | - | 35.4 | 39.1 | 37.2 | 9.6 | - | 33.4 | 36.7 | 34.9 |
| | NIR | 10.2 | 93.5 | 45.1 | 54.2 | 49.2 | 11.0 | 95.5 | 44.9 | 43.2 | 44.0 | 10.6 | 96.0 | 34.1 | 40.9 | 37.2 |
| | FR* | 9.0 | 93.5 | 55.5 | 58.9 | 57.1 | 11.5 | 94.5 | 44.8 | 44.7 | 44.8 | 11.0 | 96.0 | 34.9 | 43.4 | 38.7 |
| | FR+A2I | 9.9 | 94.0 | 53.2 | 62.1 | **57.3** | 11.5 | 97.0 | 47.7 | 47.7 | **47.7** | 10.8 | 95.5 | 35.9 | 43.7 | **39.4** |

Inter_RAT and FR, Table 1 and 2), bert-base-uncased (following CR, Table 4), and GCN (for the BA2Motifs dataset). The random seed is kept the same (the seed is 12252018, inherited from the code of FR) across all the experiments on text classification, as we think experiments with multiple datasets and multiple sparsity settings (totally 12 settings in Table 1 and 2) under the same random seed are sufficient to verify the significance of improvement. For the BA2Motifs, we use a two-layer GCN. The training of GCN is not as stable as GRUs, and we report the average results of five random seeds. More details are in Appendix A.5.

## 5.1 RESULTS

**Rationale quality**. Table 1 and 2 show the results on the text classification datasets. For the most widely used beer-related datasets (which have been the most important benchmarks for a long time), we follow Inter_RAT to set three different sparsity levels: $10\%, 20\%, 30\%$, by adjusting $s$ in Equation (4). For the hotel-related datasets, we use them as supplementary material and follow FR to set the sparsity to be

Table 3: Results on BA2Motifs. "()": std.

| Methods | S | Acc | P | R | F1 |
|---|---|---|---|---|---|
| Comparison with standard RNP | | | | | |
| RNP | 20.3 (2.5) | 95.2 (1.9) | 36.5 (5.5) | 36.5 (2.2) | 36.4 (3.8) |
| RNP+A2I | 20.5 (2.3) | 95.2 (1.5) | 39.7 (3.5) | 40.5 (2.9) | **40.0** (2.5) |
| Comparison with advanced variants | | | | | |
| FR | 20.5 (2.3) | 96.4 (1.8) | 39.3 (5.9) | 40.0 (4.9) | 39.6 (5.2) |
| FR+A2I | 20.2 (1.5) | 96.5 (1.4) | 42.1 (2.8) | 42.5 (4.0) | **42.3** (3.0) |

similar to human-annotated rationales. Initially, we conduct our attacking inspection on top of the standard RNP to validate our claims and demonstrate the efficacy of our proposed method. Across all nine settings in Table 1, we observe a significant improvement over the standard RNP in terms of F1 score. Notably, the highest increase reaches up to $9.0\%$ (*Beer-Appearance* with $S \approx 10\%$), underscoring the robust effectiveness of our method. Additionally, we compare with a representa-

Table 4: Results with BERT. We follow CR to set $S \approx 10\%$. "*": results obtained from CR.

| Datasets | | Beer-Appearance | | | Beer-Aroma | | | Beer-Palate | | |
|---|---|---|---|---|---|---|---|---|---|---|
| Methods | | P | R | F1 | P | R | F1 | P | R | F1 |
| $S \approx 10\%$ | RNP* (Lei et al., 2016) | 48.7 | 11.7 | 20.0 | 44.2 | 20.7 | 27.6 | 25.1 | 21.9 | 22.8 |
| | A2R* (Yu et al., 2021) | 49.1 | 18.9 | 25.9 | 51.2 | 21.2 | 29.8 | 31.8 | 24.3 | 25.4 |
| | CR* (Zhang et al., 2023) | 45.3 | 22.0 | 28.0 | 60.3 | 35.4 | 39.0 | 32.5 | 25.9 | 26.5 |
| | FR (Liu et al., 2022) | 41.8 | 19.3 | 26.4 | 47.1 | 27.6 | 34.8 | 32.6 | 29.4 | 30.9 |
| | FR+A2I | 48.6 | 25.7 | **33.6** | 55.4 | 32.0 | **40.5** | 34.4 | 32.3 | **33.3** |

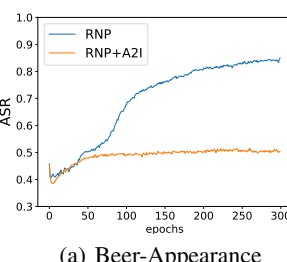
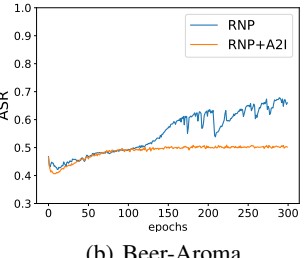
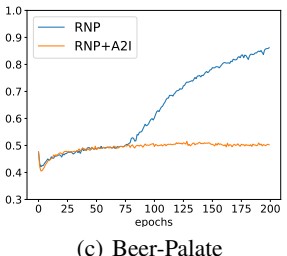

(a) Beer-Appearance        (b) Beer-Aroma        (c) Beer-Palate

Figure 6: Attack success rate on the three beer-related datasets. The rationale sparsity is about $20\%$.

tive LLM, llama-3.1-8b-instruct in Table 6 of Appendix A.6, and find that our simple A2I-based methods get comparable results to it and can sometimes even outperform it.

Our attack-based inspection is more of a tool than an independent model and is model-agnostic (as long as there is a predictor to attack). Therefore, we further apply it on top of the advanced method, FR (as FR outperforms Inter_RAT and NIR in most cases), to demonstrate our competitiveness. Two observations emerge from the results. When our A2I is incorporated, the performance of both RNP and FR consistently improves. We observe a significant improvement in FR's performance (up to $6.9\%$ on *Beer-Appearance* with $S \approx 10\%$) when our A2I is layered atop it, highlighting the competitiveness of our method. Aside from the most widely used beer-related datasets, we also consistently achieve strong performance on the hotel-related datasets and the graph dataset BA2Motifs (note that Inter_RAT, NIR, and CR are methods specifically designed for text tasks and are not suitable for graph tasks).

**Results with BERT**. To show the competitiveness of A2I, we also follow CR to conduct experiments with pretrained BERT on the three most widely used beer-related datasets (Table 4) and compare with some methods that have already been implemented with BERT. We still get considerable improvements as compared to recent methods.

**Attack Success Rate (ASR)**. To more effectively demonstrate the capabilities of our attacking inspection, we present the attack success rates for both RNP and our RNP+A2I. This experiment aims to address two key questions: 1) Can the attacker truly identify the trivial patterns recognized by the predictor? 2) Can the inspection really prevent the predictor from adopting the trivial patterns? ASR is a metric commonly employed in the realm of security. Given a pair $(X, Y)$, if $f_p(f_a(X)) = 1 - Y$, indicating a label inversion, we deem the attack successful. ASR serves as an indicator of both an attack method's efficacy and a model's resilience against such attacks. A high ASR signifies the effectiveness of an attack method, while a low ASR denotes model robustness. The results for the three beer-related datasets are displayed in Figure 6. Regarding the first question, "Can the attacker truly identify the trivial patterns learned by the predictor?", the blue lines offer insight. As opposed to RNP+A2I, the blue lines depict models where we omit the objective Equation (11) (specifically, the instruction loss) from Equation (7). This means that while RNP is trained as usual, an attacker is also being trained concurrently. The prominence of the blue lines demonstrates that the attacker achieves a remarkably high ASR. This indicates that the predictor in RNP does internalize some trivial patterns, and the attacker successfully identifies them, underscoring the potency of the attack. For the second question, "Can the inspection effectively deter the predictor from adopting trivial patterns?", we can look to the orange lines. The ASR values hover around $50\%$, which is close to random classification. This suggests that the attacker can only select some neutral patterns and the predictor actively avoids learning from the trivial patterns, highlighting the efficacy of the instruction.

## 6 Conclusion and limitations

This paper investigates a new type of spurious correlation (i.e., model-added spurious correlation) in the self-explaining rationalization framework. It can appear even in clean datasets, thus making previous causal methods (which focus solely on the causal relationships in the raw dataset) ineffective in dealing with it. We design an attack-based method to inspect the model-added spurious correlations and to instruct the training of rationalization. Experiments on six text classification datasets and one graph classification dataset show the effectiveness of the proposed method.

One limitation is that although we have provided the method for n-class classification, the experiments are conducted on binary classification datasets. This is because there are no proper multi-class classification datasets that contain ground-truth rationales (as it usually requires more domain expertise to annotate rationales than to annotate the class label) for evaluation. In the future, we will consider seeking more collaborators to create better benchmarks.

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

Table 5: Statistics of datasets used in this paper

| Datasets | | Train | | | Dev | | | Annotation | | | |
|---|---|---|---|---|---|---|---|---|---|---|---|
| | | Pos | Neg | avg_len | Pos | Neg | avg_len | Pos | Neg | avg_len | S |
| Beer | Appearance | 16891 | 16891 | 141 | 6628 | 2103 | 145 | 923 | 13 | 126 | 18.5 |
| | Aroma | 15169 | 15169 | 144 | 6579 | 2218 | 147 | 848 | 29 | 127 | 15.6 |
| | Palate | 13652 | 13652 | 147 | 6740 | 2000 | 149 | 785 | 20 | 128 | 12.4 |
| Hotel | Location | 7236 | 7236 | 151 | 906 | 906 | 152 | 104 | 96 | 155 | 8.5 |
| | Service | 50742 | 50742 | 154 | 6344 | 6344 | 153 | 101 | 99 | 152 | 11.5 |
| | Cleanliness | 75049 | 75049 | 144 | 9382 | 9382 | 144 | 99 | 101 | 147 | 8.9 |

# A    EAXAMPLES AND IMPLEMENTATIONS DETAILS

## A.1    THE POTENTIAL IMPACT OF RATIONALIZATION IN THE ERA OF LLMS

In comparison to traditional "model-centric" XAI methods which solely focus on the model's learned information, "data-centric" approaches primarily aim to extract model-agnostic patterns inherent in the data. So, apart from improving interpretability, rationalization can serve as a method of data cleaning (Seiler, 2023).

Domain-specific large models often require supervised fine-tuning using domain-specific data. Uncleaned data may contain harmful information such as biases and stereotypes (Sun et al., 2024). Recent research suggests that training predictors with extracted rationales can remove irrelevant harmful information, enhancing robustness (Chen et al., 2022) and generalization (Wu et al., 2022; Gui et al., 2023).

Since LLMs are usually pretrained on various datasets, they tend to be less controllable than small models (Zhao et al., 2023). Considering that for simple tasks (such as text classification), small models are also capable and can achieve satisfactory results, we can train a separate rationalization model for a single domain-specific dataset. Small models trained on a single dataset are often more controllable and save computational resources (such as searching for hyperparameters and adding regularization terms) (Guo et al., 2023). Then using the extracted rationales for supervised fine-tuning might prevent large models from learning harmful information from new data. Additionally, shortening input texts can also reduce the memory required for fine-tuning.

A recent study has also found that training a small model for data selection (although not the same as rationale selection) and producing a small subset is useful for fine-tuning LLMs (Xia et al., 2024).

## A.2    A TOY EXAMPLE FOR A MORE INTUITIVE UNDERSTANDING OF THE PROPOSED METHOD

Firstly, to inspect and identify the correlations, we introduce an attack generator $g_a$. Figure 7 shows an example of how the attacker works (formal analysis is in §4.3).

*Example 2*: the optimization objective of $g_a$ is to select an attack rationale $Z_A$ from input such that, when $Z_A$ is fed into the same predictor $p$, it yields a prediction label flipped from its original label. Continuing the previous example in Figure 2, the generator $g$ selects the "." from a positive input $X^1$ with label 1 as $Z$. Consequently, the predictor $p$ learns to treat the presence of "." in $Z$ as an indicative feature for positive classification. On the other hand, the goal of $g_a$ is to select an attack rationale $Z_A$ from a negative input $X^0$ with a label 0 in such a way that, when $Z_A$ is fed to the same predictor $p$, the prediction result flips from its original label 0 to 1. Achieving this objective is straightforward: $g_a$ simply needs to mimic $g$ by selecting "." as $Z_A$. This suggests that if $g$ identifies $Z$ from $X^1$ as a trivial pattern also present in $X^0$, then $g_a$ can effortlessly select $Z_A = Z$ from $X^0$, leading to an easy flip of the prediction label of $Z_A$ to 1 in predictor $p$. On the other hand, if $Z$ is a genuine positive rationale unique to $X^1$ and the predictor $p$ classifies it correctly, then $g_a$ would be unable to find a positive rationale from the negative input $X^0$. Therefore, it is difficult for the predictor $p$ to flip $Z_A$'s label from 0 to 1. Thus, we can leverage the attack generator $g_a$ to assist in inspecting and identifying sampling bias. $g_a$ may easily find a $Z_A$ that flips its predicted label in predictor $p$ from its actual label, indicating the presence of semantically unrelated trivial patterns in $Z$.

To further address this issue, we propose a method to instruct the game on better decorrelation. As illustrated by the previous example, when there is a sampling bias issue, the attack generator $g_a$ surely selects a $Z_A$ that is a trivial pattern lacking semantic significance. For a reasonable predictor $p$ that can accurately classify the real rationale, $Z_A$ is akin to noise, and its classification result should be random and not biased towards any label. Therefore, we introduce a constraint on the predictor $p$ to guide it, ensuring that the classification result for $Z_A$ remains as random as possible. This constraint serves as an ongoing guidance to adjust and correct the behavior of predictor $p$. An improved predictor $p$ can, in turn, better instruct and guide the updates for the generator $g$.

## A.3 IMPLEMENTATION DETAILS OF EQUATION (6) AND (7)

For a batch of $(X, Y)$, we first send $X$ to both the generator and the attacker and get $Z, Z_A$:

$$Z = f_g(X)$$
$$Z_A = f_a(X). \tag{12}$$

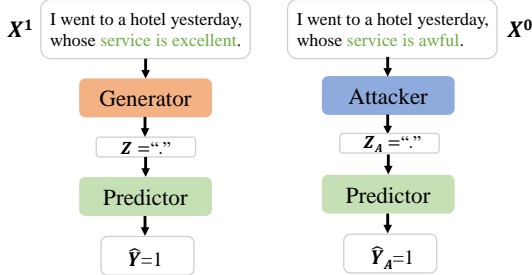

Then, we get a copy of $Z_A$ with the pytorch function "torch.detach()":

$$Z'_A = \text{torch.detach}(Z_A). \tag{13}$$

Then we get $\hat{Y}$ and $\hat{Y}'_A$:

$$\hat{Y} = f_p(Z)$$
$$\hat{Y}'_A = f_p(Z'_A) \tag{14}$$

Figure 7: An example of how the attacker works. $X^1, X^0$ represent positive and negative texts.

Then we can update the generator and the predictor with

$$\min_{\theta_g, \theta_p} H_c(Y, \hat{Y}) + \min_{\theta_p} H_c([0.5, 0.5], \hat{Y}'_A) \tag{15}$$

Note that this updating process will not influence the attacker, since we have used "torch.detach()" for $Z_A$.

Then, we fix the parameters of the generator and the predictor, and only update the attacker. We get $\hat{Y}_A$ with

$$\hat{Y}_A = f_p(Z_A). \tag{16}$$

Then, we update the attacker with

$$\min_{\theta_a} H_c(1 - Y, \hat{Y}_A). \tag{17}$$

Then, we get into the next round to update the generator and the predictor again.

## A.4 DATASETS

We employ six widely used text classification datasets collected from two rationalization benchmarks. Beer-Appearance, Beer-Aroma, Beer-Palate (which discuss the appearance, aroma, and palate of beer, respectively. They are from the BeerAdvocate (McAuley et al., 2012) benchmark), Hotel-Location, Hotel-Service, Hotel-Cleanliness (which discuss the location, service, and cleanliness of hotels, respectively. They are from the HotelReviews (Wang et al., 2010) benchmark). Among them, the beer-related datasets are most important and used by nearly all of previous research in the field of rationalization. These datasets have human-annotated ground-truth rationales on the test sets for evaluation. But the training sets have only the classification labels and models are trained to extract rationales in an unsupervised way.

For the three beer-related datasets, users need to consult the original authors (McAuley et al., 2012) for permission first.

The statistics of the datasets are in Table 5. $Pos$ and $Neg$ denote the number of positive and negative examples in each set. $S$ denotes the average percentage of tokens in human-annotated rationales to the whole texts. $avg\_len$ denotes the average length of a text sequence.

Note that there are two versions of the BeerAdvocate benchmark. The raw datasets in the original BeerAdvocate contain many spurious correlations. However, as we are investigating the model-added spurious correlations in clean datasets, we follow FR to use the version where the inherent spurious correlations in the datasets have been manually cleaned by Lei et al. (2016).

For the graph classification dataset BA2Motif, we do node level selection on it. That is to say, we select several nodes from a graph to form a subgraph to serve as the rationale.

## A.5 Implementation details

We keep the major settings consistent with Inter_RAT and FR, which are commonly utilized in the field of rationalization (Chang et al., 2020; Yu et al., 2021; Liu et al., 2022; Yue et al., 2023). Specifically, we employ the 100-dimensional GloVe (Pennington et al., 2014) for word embedding and 200-dimensional GRUs (Cho et al., 2014) to obtain text representation. The re-parameterization trick for binarized selection is Gumbel-softmax (Jang et al., 2017). Then, we also follow CR to conduct experiments that replace GRUs with pretrained BERT (Devlin et al., 2019) ("bert-based-uncased") and compare with some recent methods that have already been implemented with BERT as a supplement. The random seed is kept the same (the seed is 12252018, inherited from the code of FR) across all the experiments on text classification, as we think experiments with multiple datasets and multiple sparsity settings (totally 12 settings in Table 1 and 2) under the same random seed are sufficient to verify the significance of improvement. For the BA2Motifs, we use a two-layer GCN to replace GRUs. The training of GCN is not as stable as GRUs, we report the average results of five random seeds.

Because Inter_RAT, NIR, and CR are methods specifically designed for text tasks and are not suitable for graph tasks, we only compare our A2I with RNP and FR on the BA2Motifs dataset.

The maximum sequence length is set to 256. We use the Adam optimizer (Kingma & Ba, 2015) with its default parameters, except for the learning rate (the learning rate is 0.0001). The temperature for gumbel-softmax is the default value 1. We implement the code with Pytorch on a RTX3090 GPU.

**Hyperparameters**. For all datasets, we use a learning rate of 0.0001. The batchsize is 128 for the beer-related datasets and 256 for the hotel-related datasets. These hyperparameters are found by manually tune the standard RNP and are applied to both NIR, FR, our A2I, as they are all variants of RNP. The core idea of NIR is to inject noise into the selected rationales. We use RNP as its backbone. A unique hyperparameter of NIR is the proportion of noise. Following the method in the original paper, we searched within $[0.1, 0.2, 0.3]$ and found that $0.1$ yielded the best results on most datasets, hence we adopted $0.1$ for it. We found that the training of Inter_RAT is very unstable. To avoid potential unfair factors, our main settings are determined with reference to it. Except for the part about sparsity, we used its original hyperparameters for it.

For CR, we just keep the major settings ("bert-base-uncased", the Beer-Appearance dataset, and the sprasity of $10\%$, removing the coherence regularizer) the same as it and copy its results from its original paper.

## A.6 The rationales extracted by llama-3.1-8b-instruct

To further show the potential impact of rationalization in the era of LLMs, here we present the results of the experiments conducted with the llama-3.1-8b-instruct model. We perform both 2-shot prompting and supervised fine-tuning.

For 2-shot prompting, we provide the model with a negative text with its corresponding ra-

> **Task:** Sentiment classification about Beer's appearance
> **Input:** Pours a rather crisp yellow almost orange with a thin head. The aroma is dominated by sweet malts with just a slight hoppiness dancing in the background. The taste does have a surprising amount of hoppiness for a Pilsner. There is a good maltiness to it as well, but citrus hops just slightly overpower. The beer is very light and refreshing. This makes for an excellent summer session beer.
> **Expected output:** 1|pours a rather crisp yellow almost orange with a thin head .
> **llama-3.1 output:** 1|pours a rather crisp yellow almost orange

Figure 8: An example of llama's output. Here "1" means that the class label $Y$ is positive. And the words after "|" represent the rationale.

tionale, and a positive text with its corresponding rationale. For supervised fine-tuning, the supervison label is the classification label, since we perform unsupervised rationale extraction. We use 4*RTX 4090 24GB GPUs and LoRA to fine tune the models. We provide a detailed document in

Table 6: The comparison between our A2I-based methods (implemented with GRUs, which corresponds to the results in Table 1) and a representative LLM llama-3.1-8b-instruct. The **bold** results means the situations where A2I-based methods outperforms llama (in terms of F1 score).

(a) Results on datasets from the BeerAdvocate benchmark.

| Datasets | Beer-Appearance | | | | Beer-Aroma | | | | Beer-Palate | | | |
|---|---|---|---|---|---|---|---|---|---|---|---|---|
| Methods | S | P | R | F1 | S | P | R | F1 | S | P | R | F1 |
| llama (finetune) | n/a | 86.3 | 46.2 | 60.2 | n/a | 73.2 | 50.6 | 59.8 | n/a | 61.7 | 42.6 | 50.4 |
| llama (2 shot) | n/a | 15.4 | 16.0 | 15.7 | n/a | 17.9 | 24.2 | 20.6 | n/a | 13.0 | 22.2 | 16.4 |
| RNP+A2I | 10.8 | 78.3 | 45.8 | 57.8 | 9.8 | 86.0 | 54.3 | **66.6** | 10.9 | 66.3 | 58.2 | **62.0** |
| FR+A2I | 11.3 | 76.0 | 46.5 | 57.7 | 10.0 | 85.7 | 54.8 | **66.9** | 9.7 | 71.4 | 55.8 | **62.6** |
| RNP+A2I | 20.0 | 73.3 | 79.4 | **76.2** | 19.5 | 49.0 | 61.4 | 54.5 | 19.4 | 49.0 | 76.4 | **59.7** |
| FR+A2I | 19.8 | 80.0 | 85.6 | **82.7** | 19.4 | 64.2 | 80.0 | **71.2** | 19.2 | 44.2 | 68.2 | **53.7** |
| RNP+A2I | 29.9 | 59.3 | 95.9 | **73.3** | 27.8 | 44.5 | 79.3 | 57.0 | 30.5 | 30.8 | 75.5 | 43.7 |
| FR+A2I | 28.8 | 61.3 | 95.3 | **74.6** | 30.9 | 41.4 | 82.2 | 55.1 | 29.1 | 31.6 | 73.8 | 44.2 |

(b) Results on datasets from the HotelReview benchmark.

| Datasets | Hotel-Location | | | | Hotel-Service | | | | Hotel-Cleanliness | | | |
|---|---|---|---|---|---|---|---|---|---|---|---|---|
| Methods | S | P | R | F1 | S | P | R | F1 | S | P | R | F1 |
| llama-3.1-8b (finetune) | n/a | 58.6 | 39.0 | 46.8 | n/a | 77.3 | 40.6 | 53.3 | n/a | 54.9 | 31.3 | 39.9 |
| llama-3.1-8b (2 shot) | n/a | 45.8 | 59.1 | 51.6 | n/a | 45.3 | 51.7 | 48.3 | n/a | 39.3 | 43.0 | 41.1 |
| RNP+A2I | 9.0 | 50.2 | 53.4 | **51.7** | 11.6 | 46.8 | 47.4 | 47.1 | 9.7 | 34.7 | 38.2 | 36.4 |
| FR+A2I | 9.9 | 53.2 | 62.1 | **57.3** | 11.5 | 47.7 | 47.7 | 47.7 | 10.8 | 35.9 | 43.7 | 39.4 |

our anonymous code repository (`https://anonymous.4open.science/r/A2I-A700/details_of_llms.pdf`) to include all the details (including the prompt templates, LoRA fine-tuning parameter settings, and more).

In most cases, the model can output the rationale in the correct format. Figure 8 shows an example. But in 2-shot prompting, the model sometimes outputs additional parts along with the rationale (through manual observation, this situation does not occur frequently.). Figure 9 is another example. In such cases, we use gpt-3.5-turbo to extract the content within the quotation marks.

The results are shown in Table 6. LLMs are not good at counting, so we did not constrain the percentage length (i.e., sparsity) of the rationale extracted by the model. Comparing the results of the supervised fine-tuned llama-3.1 with our results in Table 1, llama-3.1 does not have a crushing advantage. For example, on the Beer-Aroma dataset, FR+A2I outperforms llama-3.1 at sparsity levels of 10% and 20%. Similarly, on the Beer-Palate dataset, RNP+A2I also outperforms llama-3.1 at sparsity levels of 10% and 20%. Besides, our A2I can be applied to graph data, while it is not easy to do so for LLMs.

# B    TECHNICAL PROOFS

## B.1    DERIVATION OF EQUATION (9)

To begin with, we need to introduce two fundamental properties from probability theory.

The first property is a general property for conditional probability. If $0 < P(Y = 1) < 1$, then for $\forall p$, if $0 < p < 1$, we can always find a variable $c$, such that $P(Y = 1|c) = p$.

Considering our rationalization situation, we can get the following corollary:

**Corollary 1.** *If we can construct $G$ in an arbitrary way, and $0 < P(Y = 1|Z = t) < 1$, then we have*

$$\forall 0 < p < 1, \ \exists g_a \in G, \ P(Y = 1|Z = t, g_a) = p. \tag{18}$$

The second property is also a general property for conditional probability. If $P(Y = 1) = 0$, then for any variable $c$, we always have $P(Y = 1|c) = 0$. This is also a fundamental property in probability theory.

Considering the rationalization situation, let $Z = r_+$, we have

**Corollary 2.** *If we can construct $G$ in an arbitrary way, and $P(Y = 0|Z = r_+) = 0$, then we have that there is no $g_a \in G$ that can make $P(Y = 1|Z = r_+, g_a) > 0$.*

### B.2 THE CONVERGENCE OF EQUATION (7)

**Quantitative analysis** We consider the scenario where the model is functioning correctly, meaning the predictor classifies according to the true rationale $R$. The generator extracts $r_+$ from $X^1$, while the attacker extracts $r_+$ from $X^0$ ($X^0, X^1$ denote texts with negative and positive labels, respectively). In the dataset $\mathcal{D}$, we consider the numbers of positive and negative texts are both $n$.

We only consider $Z = r_+$, and $Z = r_-$ is nothing different.

We rewrite (7) as ($L$ is the loss function and $f_p(r_+)$ is the confidence level of predicting $r_+$ as positive)

$$
L = -\sum_{Y=1,X} \mathbb{1}_{f_g(X=r_+)} \log f_p(r_+)
$$
$$
-\sum_{Y=0,X} \mathbb{1}_{f_a(X=r_+)} 0.5\left(\log f_p(r_+) + \log(1 - f_p(r_+))\right)
$$
$$(19)$$

> **Task:** Sentiment classification about Beer's appearance
> **Input:** 22oz bottle pouted into a Goblet: Opaque orange with a light, white, creamy head that was not all that well retained but full of carbonation, but did settle into a small thin cap. The aroma was more Belgian triple than IPA, Sweet and malty. The taste is a very nice balance of the two styles. A little more hops, but balanced very nice with the sweetness of the malt and fruit. The beer had a medium to full body, perhaps a little too thick for my taste, but still good. The beer had a nice bitter dry aftertaste and was well carbonated. The beer was fairly easy to drink give the ABV, but after the 22oz, I was pretty well done. Overall, a good beer and probably the first one of the side projects that I think the brewery should consider brewing on a regular basis.
> **Expected output:** 1|opaque orange with a light , white , creamy head that was not all that well retained but full of carbonation , but did settle into a small thin cap .
> **llama-3.1 output:** positive|The overall tone of the review is positive, with phrases such as "a very nice balance of the two styles", "nice bitter dry aftertaste", "well carbonated", and "overall, a good beer" indicating a favorable opinion of the beer.
> **GPT-3.5-restructured output:** 1|a very nice balance of the two styles nice bitter dry aftertaste well carbonated overall, a good beer

Figure 9: An example of llama fails to output the rationale in the right format.

$$
\frac{\partial L}{\partial f_p(r_+)} = \frac{-n * \Pr(r_+|Y = 1) - 0.5n * \Pr(r_+|Y = 0)}{f_p(r_+)}
$$
$$
+ \frac{0.5n * \Pr(r_+|Y = 0)}{1 - f_p(r+)}
$$
$$(20)$$

We consider a scenario starting with $f_p(r_+) = 0.5$, meaning the predictor is unable to classify using the correct rationale, and we examine in which direction the predictor will converge under these circumstances.

Clearly, when $f_p(r_+) = 0.5$, $\frac{\partial L}{\partial f_p(r_+)} < 0$, meaning that the predictor will learn to increase $f_p(r_+)$ to get lower $L$. So the predictor will learn to predict $r_+$ as positive.

So, when will it converge? We denote $\Pr(r_+|Y = 1) = P_1$ and $\Pr(r_+|Y = 0) = P_2$. From (20), we have

$$
\frac{\partial L}{\partial f_p(r_+)} < 0, \ s.t., \ f_p(r_+) < 1 - \frac{P_2}{2P_1 + 2P_2}.
$$
$$(21)$$

From Assumption 1, we have $P_1 \geq P_2$. So, we know that we will have $f_p(r_+) \geq 0.75$ when the predictor converges (i.e., $\frac{\partial L}{\partial f_p(r_+)} = 0$).

That means even in the worst case, the predictor can still predict $r_+$ as positive.

**Qualitative analysis** Actual training would be easier because, in the above discussion, we do not differentiate between positive sentiment appearing in positive class texts and positive sentiment ap-

pearing in negative class texts. In reality, although both are denoted as $r_+$, they are somewhat distinct.

Here are some practical scenarios where a text contains both positive and negative sentiments.

First, the $X$ labelled with $Y = 1$ may be a combination of strong positive sentiment and weak negative sentiment. A dataset may consists of two kind of sentiment: strong and weak, each of which can be divided to positive and negative. The label of $X$ is decided by the strong sentiment. In this scenario, the attacker may find the weak negative sentiment from $X$ labelled with $Y = 1$, and ask the predictor to classify the weak negative sentiment as neutral. If weak sentiment and strong sentiment have different styles, the attacker here still helps the predictor to focus on strong sentiment and ignore the weak sentiment. As a result, the generator will only select the strong sentiment.

Second, the sentiment may be multi-aspect. For example, a person may have positive sentiment about the beer's appearance, while negative sentiment about the taste. If we are discussing the beer's appearance, the text will still be annotated as positive. In such a scenario, the attacker will try to find the negative comment about the taste, and force the predictor to classify it as neutral. However, this is just what we want. It helps the predictor focus not only on the vanilla sentiment, but also on the aspect (which is included in the context of the sentiment) in which we are interested. Since the predictor classifies the comment about the taste as neutral, it will give the only the feedback about the beer's appearance, which can help the generator focus more on the appearance.

The above intuitive analysis is somewhat supported by the empirical results in Figure 6. For RNP+A2I, the attack success rate is about $50\%$, meaning random classification of $Z_A$. This suggests that the predictor does not predict the $Z_A$ extracted by the attacker to the target class.

### B.3    THE MINIMUM CROSS-ENTROPY IS EQUAL TO ENTROPY

$$H_c(Y, \hat{Y}|Z) = H(Y|Z) + D_{KL}(P(Y|Z)\|P(\hat{Y}|Z)). \tag{22}$$

We have $D_{KL}(P(Y|Z)\|P(\hat{Y}|Z)) \geq 0$ with the equality holds if and only if $P(Y|Z) = P(\hat{Y}|Z)$. As a result, we have

$$\min H_c(Y, \hat{Y}|Z) = H(Y|Z). \tag{23}$$

