# OpenReview forum: "Attacking for Inspection and Instruction: Attack Techniques Can Aid In Interpretability"
_ICLR.cc/2025/Conference — ICLR 2025 Conference Withdrawn Submission_

### Official Review · Reviewer_SpHe · 2024-10-30

**Soundness:** 3
**Presentation:** 1
**Contribution:** 2
**Rating:** 3
**Confidence:** 3

**Summary:**

This paper introduces a new component into the framework of Rationalizing Neural Predictions (Lei et al., 2016), which facilitates interpretable (text) classification. RNPs consist of a generator that selects a part of an input and a predictor that uses this part to classify a label. The first contribution is highlighting that a generator can introduce spurious correlations into the predictor. Second, an attacker network that identifies potential spurious correlations in data and then instructs the predictor not to learn from these trivial patterns is introduced. A well-instructed predictor can give good feedback to the generator’s selection, which improves the model's robustness / generalization / rationale capabilities.

**Strengths:**

1. The main strength of the paper lies in its **first contribution** explained across Sections 1 & 4.1 (+Appendix). Highlighting the new type of spurious correlation stemming from conditioning on the generator is interesting and significant. Knowledge of quite simple statistics facts on conditional independence leads to correcting a rather empirical line of research on interpretable NLP.
2. I like all the figures and tables in the paper.

**Weaknesses:**

1. **Idea:** I am not convinced that the **second contribution**, i.e. a practical solution using an attacker, is useful. This paper might try to solve a problem that never even existed in a similar parallel line of work on interpretable (text) classification. What is the, unmentioned in the paper, relation of RNPs to Prototype-based Networks? E.g.
    * Interpretable and steerable sequence learning via prototypes, KDD 2019
    * Evaluating explainable AI: Which algorithmic explanations help users predict model behavior?, ACL 2020
    * ProtoTEx: Explaining model decisions with prototype tensors, ACL Main 2022
    * Proto-lm: A prototypical network-based framework for built-in interpretability in large language models, EMNLP Findings 2023
    * ProtoryNet - Interpretable text classification via prototype trajectories, JMLR 2023
    * Robust text classification: Analyzing prototype-based networks, EMNLP Findings 2024

**Sidenote:** I peeked inside the related papers on "Inter RAT (Yue et al., 2023), CR (Zhang et al., 2023), FR (Liu et al., 2022) and NIR (Storek et al., 2023)." None of them ever mention prototype-based networks (related to  "This looks like that ..." NeurIPS 2019) or concept bottleneck models (related to "Concept bottleneck models" ICML 2020), which seems like a systemic weakness of work on RNPs.

2. **Experiments:**
    * A) Contain no comparisons to popular methods in text classification, e.g. fastText (Bag of tricks for efficient text classification, EACL 2017), or even baseline BERT (without RNP) etc. It is valuable to compare with uninterpretable models to observe the baseline / tradeoff for a broader context on the progress in the field.
    * B) There is no analysis of the method's efficiency. One can expect to experience the computational drawbacks of training the attacker. How much overhead is added by adding an attacker? How does the convergence / learning speed / sample efficiency compare to the related baselines?

3. **Communicaton:** I believe the paper's title is *very* uninformative considering the paper's contents, to the point that I want to raise this as an issue. First of all, from its beginning, the paper assumes a particular modeling framework called *Rationalizing* Neural Predictions, which is primarily used in *text* / *language* modeling. Both concepts are not clearly represented in the title or abstract, where the two first sentences read more like the paper itself introduces an idea of RNPs. Moreover, the title misses a critical concept discussed throughout the paper, i.e. *spurious correlation*. Emphasizing "attack" two times may be misleading for those who actually work on adversarial ML. "Interpretability" seems vague as both *robustness* (generalization) and *causality* are even more emphasized in the paper. The word "techniques" is uninformative (which techniques? one or many attacks?) and is not even used later in the paper.

**Questions:**

1. I can suggest emphasizing *discrimination* (cf. GANs) instead of "attacking" across the paper for clarity since no attacking entity is considered in this work. "An attacker" is confusing in the context of actually improving interpretability / robustness.
2. Figure 4: Using the accuracy metric may be misleading as there is no information regarding the class (im)balance. Please either plot a horizontal line showing the class ratio, comment on it in the figure's caption, or plot F1 etc.
3. Assumption 1 becomes unrealistic in scenarios with multiple classes. Binary sentiment classification is an oversimplified example.
4. Overall, I like Section 4.2. But, I disagree with the rationale given in L262–270 regarding the result in Figure 4:
> "Does this strange result stem from the fact that the 10% randomly selected patterns already contain enough sentiment inclination for classification? The answer is no. [...] We observe that the green line indicates a significantly lower accuracy (about 58%), implying that the randomly selected patterns contain only minimal sentiment information."

In my opinion, the answer is we don't know:
* A) It is probable that the predictor trained using the full texts (green line) itself learns spurious correlations (shortcuts) that are different from these contained in the 10% randomly selected patterns.
* B) It is probable that the predictor learns variable interactions, e.g. one important word lies inside the 10%, and another one lies inside the 90%; access to both is required for accurate prediction (rationale).

Thus, the implication seems incorrect.

### Other feedback
- L36: introduce the abbreviation for "XAI"
- L83–85: "This phenomenon then leads to a trust concern: whether the extracted rationale is really responsible for the label in the original dataset. This problem is important because explanations should also be aligned with their social attribution (Jacovi & Goldberg, 2020; 2021)." I can disagree; explanations don't have to be aligned with their social attribution but rather be faithful to the model. Confirmation bias is a real threat to progress in research on interpretability.
- RW: Authors might be interested in a very related work: "Post hoc explanations may be ineffective for detecting unknown spurious correlation" ICLR 2022
- L92: What is denoted by the letter "g"; generator? It was never introduced.
- L127: typo, missing reference
- L193: typo, methods constrain
- L232: please rephrase "it will *sometimes* results in *some* problems."
- L242/Fig.3: "a local *of* the causal graph" sounds odd
- Eq. 8: missing spaces next to " & "
- L281: use another letter instead of "n", which was used before to denote the number of variables (T_1, ..., T_n)
- Figure 5: wrong wording in "Attack *to* Inspection and Instruction", do you mean "for" or "as"?
- L310: "Inspection" seems to be a new concept introduced here, but the paper's Introduction gives no intuition of what it really means "to inspect". Also, "the trivial patterns learned by the predictor can be inspected through attack" sounds like defining "inspection" by using the word "inspect". I like the sentence in L331 explaining that "an attacker can identify uninformative trivial patterns and classify them into the opposite class." and can recommend moving this explanation to the beginning of Sec. 4.3 or even Introduction.
- L342: wrong wording in "The situation of a text X contains", do you mean "if", "when", or "containing"?

---

### Official Review · Reviewer_WyTS · 2024-10-31

**Soundness:** 2
**Presentation:** 3
**Contribution:** 2
**Rating:** 3
**Confidence:** 4

**Summary:**

This study investigated a self-explaining framework constructed with a cooperative game, which includes a generator and predictor for rationale discovery and tasks prediction. The authors proposed to inspect spurious correlations through attacks, and provide guidelines to stop the predictor from learning these correlations.

**Strengths:**

This study introduced a two-step method for detecting spurious correlations: inspection and instruction. The authors first developed an attacker model to identify the trivial patterns that result in spurious rationale. Subsequently, they proposed a strategy to inhibit the predictor from acquiring such correlations.

**Weaknesses:**

The authors claimed that their method distinguishes from existing causality research for spurious correlation. Instead of focusing on spurious correlations inherent in the data, they are looking for spurious correlations from the selection process of the generator. However, this claim appears ambiguous and requires further explanation. Typically, if a correlation is spurious and does not derive from the data itself, it is often due to model bias and hence, is model-specific. Yet, this paper also posits that their proposed method is model agnostic. The authors should elucidate these claims in their paper.

Considering the aforementioned concern, the contribution of this paper appears to be not significant. Furthermore, the approach of using attack techniques to identify spurious correlations has been previously explored in the literature, such as in "Mitigating Backdoor Poisoning Attacks through the Lens of Spurious Correlation" (https://arxiv.org/pdf/2305.11596).

The impact of this work is small. It is not an end-to-end tool, but instead an add-on step for existing tools.

**Questions:**

There are several grammar and format errors such as:
Line 128: ?
line 234 : "Note that independent doesn’t lead to conditional independent:"
Figure 3: "A local of the causal graph ..."

---

### Official Review · Reviewer_N2Uz · 2024-11-02

**Soundness:** 2
**Presentation:** 3
**Contribution:** 2
**Rating:** 5
**Confidence:** 3

**Summary:**

The paper studies the problem of spurious correlations introduced in self-explaining rationalizing framework. The generator that selects the relevant text for interpretability could rely on trivial and spurious text patterns when used in conjunction with a text classification model (predictor). The paper proposes an attacker to first learn a label-agnostic text selection model, which is then used to attack the predictor to check for adversarial robustness. The attacker is then used in adversarial training to minimize the reliance on the spurious text patterns and thus improves the generator/predictor accuracy as measured on 2 text classification tasks and 1 graph classification task.

**Strengths:**

* The paper studies a problem of relying on spurious text correlations in interpretability methods and demonstrates that it is an issue in existing rationalizing methods
* Experiments on a diverse set of tasks demonstrate that there is headroom in improving accuracy through an interpretable method

**Weaknesses:**

* The method is comparable to adversarial training of the classifier [1, 2, 3] where prior research has shown that adversarial training improves the robustness of the classifier. In that regard, the distinction between this work and related work in adversarial training is missing.
* The theoretical exposition is unclear - with R defined in the causal diagram but unused in the equation. The training paradigm and the intended causal diagram that is being enforced is unclear.
* Figure 2b shows that predictor trained on random selections show higher accuracy (orange) than when the predictor was trained on the full text but given random selections of text (green). Further the gap between blue and orange is only 10-15%, which indicates that the test setup is not robust enough to begin with. This has to be explained, further - in Table 1 & 2, the accuracy where any rationalization is not used, but the full text is used for classification should be provided

[1] https://aclanthology.org/P18-1099/
[2] https://dl.acm.org/doi/10.1145/3278721.3278779
[3] https://aclanthology.org/P19-1561/

**Questions:**

Discussion on the results with respect to the full-text baseline and related work in adversarial training would help increase the score.

---

### Official Review · Reviewer_EoKe · 2024-11-11

**Soundness:** 2
**Presentation:** 2
**Contribution:** 2
**Rating:** 5
**Confidence:** 1

**Summary:**

This submission examines a cooperative game framework for interpretability, where a generator identifies important parts of the input (rationale) by maximizing the mutual information with the labels, which the predictor then uses to make predictions. The authors argue that even if the predictive model achieves high accuracy, the generated rationales may be uninformative due to spurious correlations introduced during sampling. To address this, the authors propose an attacker framework that assesses whether the predictor is learning from spurious correlations and instructs the predictor to avoid such degeneracy.

---
**Disclaimer:** The topic of this paper is totally outside my area of expertise. I requested that the PCs remove me from this submission at the start of the review period but have not received a response. My evaluation below is, therefore, based only on an educated guess.

**Strengths:**

To my understanding, the proposed mechanism is both novel and interesting: spurious correlation is not an inherent property of the underlying dataset but rather an artifact of the sampling process conditioned on the generator.

**Weaknesses:**

I have the following questions and concerns:

1. I am unclear about the experimental setting in Figure 4. If the random patterns are independent of $Y$, why does the validation accuracy (orange) exceed 50%? If these random patterns do contain information relevant to the predictive task, why is the model trained on the full text unable to leverage this information effectively?

2. Section 4.2 could benefit from improved clarity. It is not immediately evident why Equation (6) suggests that the attacker "inspects trivial patterns from X." The intuition in Section 4.3 appears to be that the trivial features are those from which the classifier can also predict the opposite label. If this is the case, how does this approach provide an advantage over simply discarding features with low mutual information with the labels?

3. Within the current framework, how does the attacker differentiate patterns that might yield contrasting sentiments depending on the context? For instance, in the example in Figure 9, words like "bitter" could imply positive or negative sentiment based on surrounding context; would such patterns therefore be discarded?

**Questions:**

See Weaknesses.

---

### Note · Authors · 2024-12-07

I have read and agree with the venue's withdrawal policy on behalf of myself and my co-authors.